# OpenReview forum: "Generative Value Conflicts Reveal LLM Priorities"
_ICLR.cc/2026/Conference — ICLR 2026 Poster_

### Official Review · Reviewer_5iqJ · 2025-10-31

**Soundness:** 3
**Presentation:** 3
**Contribution:** 3
**Rating:** 8
**Confidence:** 3

**Summary:**

The paper addresses the challenge that existing LLM alignment datasets rarely contain genuine value conflicts, making it difficult to study how models behave when forced to make trade offs. CONFLICTSCOPE, the proposed system, is desigedn to generate conflict scenarios given a set of user defined values. LLMs are evaluated on open ended outputs, rather than multiple choice questions, which can be brittle. The system simulates a user prompt, and the target LLM is then prompted and its output judged against the conflicting actions. Model preferences are aggregated using a bradley terry model to produce a final ranking of the models value priorities. The paper also includes some work on steering via system prompt, to align a model to a predefined value system.

1. Scenarios generated by CONFLICTSCOPE were deemed to be more morally challenging
2. Models moved away from supporting protective values and toward supporting personal values in an open-ended eval setting.
3. Value alignment increased by an average of 14% via system prompt with value ordering

**Strengths:**

Addresses a gap in existing alignment datasets, which rarely elicit true value conflicts. The ability to generate scenarios from a set of user defined value definitions (top down) makes the approach generalizable and potentially very useful.

Similarly the focus on open ended evaluation means that the approach more closely simulates realistic LLM usage and is more reliable than multiple choice, which can be brittle and sensitive to prompt nuances and ordering effects.

Interesting experiment results.

**Weaknesses:**

The approach relies heavily on an LLM (GPT4) for filtering, simulating the user, and judging the target model's response. The filtering is validated against human judgments with high precision, but the reliance on a foundational LLM for evaluation is a dependency and introduces some bias and reproducibility questions.

**Questions:**

How would CONFLICTSCOPE work for multi turn interactions?
Did you try to mitigate for bias and reproducibility when using GPT4 as llm judge?
Do you have a theory as to why the model tend to shift from protective values to personal values in the open ended eval setting?

---

> ### Author Response · Authors · 2025-11-20
>
> We thank the reviewer for their thoughtful feedback.
>
> **Reliance on Foundation Models:** In response to reviewer concerns about the usage of a closed API model for evaluating open-ended interactions, we’ve added new experimental results with two open-weights models (Llama-3.1-70B-Instruct and Llama-3.1-8B-Instruct) to Appendix G (Robustness of Model Preferences Across Environment Model). This should make it easier for future work to replicate our usage of LLM-as-a-judge for evaluation. We discuss the results of these additional analyses below.
>
> **Multi-Turn Interactions:** In theory, it should be reasonably easy to extend our open-ended evaluation to multi-turn interactions. This could involve prompting the simulated user LLM to write followup messages, and giving the full conversation to the target LLM as we are now, until the conflict is deemed to have been resolved in a certain direction. We plan to pursue this more realistic setting in future work.
>
> **Bias Mitigation in LLM-as-a-judge:** We shared the reviewer’s concerns that our open-ended evaluation results might be sensitive to bias depending on our choice of environment model. In Appendix G of the original paper, we evaluated GPT-4.1 and Claude Sonnet 3.5 as environment models over the same set of target models and the same subset of scenarios. We computed moderate agreement (Kendall’s Tau of 0.56) between the resulting preference annotations, which is similar to the crowdworker-model IAA of 0.602. Additionally, we found that changing environment models did not cause any statistically significant changes in preference rates in any of the target models tested.
>
> In the revised version of our paper, we’ve also rerun this analysis with the open-weights models Llama-3.1-70B-Instruct and Llama-3.1-8B-Instruct. We found that when acting as the environment (judge + simulated user) model, both models have moderate agreement with GPT-4.1, with Cohen’s Kappa values of 0.506 (70B) and 0.449 (8B), respectively. Additionally, across four different environment models, the choice of environment model still did not lead to any statistically significant changes in helpfulness-harmlessness preference rate across any of the target models evaluated.
>
> **Why does the model shift across evaluation settings?:** One theory we have for the shifts across evaluation settings is that the MCQ setting more closely resembles a safety evaluation for models due to the artificial nature of the MCQ environment. This may motivate models to behave more cautiously compared to the more realistic open-ended evaluation. We hope to empirically test this hypothesis, and generally study how model behavior shifts across evaluation settings, in future work.

---

### Official Review · Reviewer_MtAF · 2025-11-01

**Soundness:** 3
**Presentation:** 3
**Contribution:** 3
**Rating:** 6
**Confidence:** 3

**Summary:**

The current work introduces an automatic pipeline by which one can conduct an evaluation of value prioritization in LLMs. The extensibility and modularity of this pipeline are what makes it shine, and resultant findings delve into the alignment issues surrounding conflicting value judgements and how LLMs act in the presence of these conflicts.

**Strengths:**

1. the motivation of all 3 research questions is well explained
2. use of multiple value sets - comprising of HHH, personal-protective, and ModelSpec - is good to ensure a holistic bucket of value set
3. Despite the use of a closed LLM for their judge, the inclusion of a human study to validate the effectiveness and accuracy of their chosen judge is a welcome sight.
4. both multiple choice and open ended evaluations are present
5. large selection of both closed and open LLMs in experimental methodology.

**Weaknesses:**

1. it would have been good to see another generative model chosen as the judge, particularly an open-weights and smaller size model

**Questions:**

1. Are there any plans to conduct a user study with the methodology to see how well it does when users supply their own value sets?

---

> ### Author Response · Authors · 2025-11-20
>
> We thank the reviewer for their thoughtful feedback.
>
> **Smaller Judge LLMs:** In response to reviewer concerns about the usage of a closed-source model for evaluating open-ended interactions, we’ve added new experimental results with two open-weights models (Llama-3.1-70B-Instruct and Llama-3.1-8B-Instruct) to Appendix G (Robustness of Model Preferences Across Environment Model). This should make it easier for future work to replicate our usage of LLM-as-a-judge for evaluation. We find that when acting as the environment (judge + simulated user) model, both models have moderate agreement with GPT-4.1, with Cohen’s Kappa values of 0.506 (70B) and 0.449 (8B), respectively. Additionally, when comparing across the four environment models, we found that changing environment models did not lead to any statistically significant changes in helpfulness-harmlessness preference rate across any of the target models evaluated.
>
> **Are there any plans to conduct a user study?:** We don’t currently have concrete plans to conduct user studies, but agree that it would be valuable to evaluate our method with user-supplied value sets.

---

### Official Review · Reviewer_bwD5 · 2025-11-01

**Soundness:** 2
**Presentation:** 2
**Contribution:** 3
**Rating:** 4
**Confidence:** 4

**Summary:**

This paper presents CONFLICTSCOPE, a generative evaluation framework that probes large language models (LLMs) under value conflict scenarios—situations where two moral or behavioral principles (e.g., autonomy vs. harmlessness) directly oppose each other. The framework automatically generates, filters such scenarios across three predefined value sets to reveal models’ implicit value priorities. Experiments on multiple LLMs show that models favor protective values in structured tasks but autonomy in open-ended ones. Furthermore, applying explicit value-order system prompts increases alignment consistency by about 14%, suggesting that value preferences can be modulated through targeted prompting.

**Strengths:**

1. Meaningful motivation.
The paper addresses an important and underexplored question, how large language models (LLMs) behave when faced with value conflicts, i.e., when two  principles (such as autonomy and harmlessness) are in opposition. This framing moves beyond standard safety or alignment metrics and provides a more realistic lens for analyzing model decision tendencies.

2. Reasonably experimental coverage.
The study includes roughly 3,000 automatically generated conflict scenarios across three distinct value domains (HHH, Personal–Protective, and ModelSpec). This scale provides sufficient empirical grounding to support its observations, with both multiple-choice and open-ended evaluations.

3. Interesting findings.
The results reveal notable behavioral patterns: models tend to prioritize protective values (harmlessness, safety) in structured multiple-choice tasks, but shift toward personal autonomy in open-ended text generation. Moreover, specifying desired value hierarchies improves alignment consistency by about 14%, suggesting that model value expression can be modulated through prompting.

**Weaknesses:**

1. Limited methodological novelty and technical depth.
The contribution lies mainly in defining a new evaluation perspective, value conflict, and constructing an automated scenario-generation pipeline. However, the framework itself relies entirely on existing LLM prompting. As a result, the work reads more as a benchmark or dataset study rather than a methodological advance suitable for a main-track research paper.

2. Lack of principled grounding for selecting value pairs.
The paper enumerates three value sets, HHH, Personal–Protective, and ModelSpec, but provides no theoretical or empirical rationale for why specific value pairs (e.g., autonomy vs. harmlessness, honesty vs. helpfulness) are chosen or considered “likely conflicts” in real-world contexts. From section 3.1, the conflict pairs are largely randomly instantiated rather than derived from sociological or psychological evidence of common moral dilemmas.

3. Reliance on LLM-as-Judge without human validation
In Section 3.1, all evaluation steps, including conflict validation, preference labeling, and response scoring are fully automated using LLM-as-judge without human verification or agreement check, which raises concerns about the objectivity and reliability of the results.

**Questions:**

1. On the selection of value conflict pairs (Section 3.1).
The author states that conflict scenarios are generated by pairing values within the three sets (HHH, Personal–Protective, ModelSpec). However, it remains unclear how these pairings were determined to represent genuine conflicts. Were they randomly combined, or was there any filtering criterion ensuring that the paired values are indeed in tension in real-world ethical reasoning.

2. On the lack of methodological clarity for target-ranking steering (Section 4).
The paper reports a “14% improvement in target ranking” with value-orderings in system prompt. However, Section 4 does not describe how this improvement was   measured, no definition of the metric, no example of the steering prompt, nor any breakdown of case studies.

---

> ### Author Response · Authors · 2025-11-20
>
> We thank the reviewer for their thoughtful feedback.
>
> **Novelty of the submitted work:** We agree that one of the main contributions of our work is motivating the need to evaluate LLM behavior under value conflict and constructing an automated data generation pipeline to do so. However, it’s not clear why this is a weakness, as ICLR’s call for papers explicitly mentions that datasets and benchmarks are appropriate subject areas for the main conference. By addressing a gap in existing alignment datasets – and a method to augment them in a scalable way – we provide a foundation for more experimental research into LLM value prioritization. Additionally, our results in Sections 5.2 and 5.3, enabled by our datasets, are also noteworthy experimental contributions in their own right.
>
> **Value Pair Selection:** We generate conflict scenarios from each pair of values in the value set, giving us exhaustive coverage of the entire value set, rather than randomly sampling a subset of pairs. We also ensure through our filtering that the generated scenarios will elicit conflict between value pairings whenever possible. As discussed in Section 2, we focus on value sets (HHH, Values in the Wild, ModelSpec) that model developers have targeted, or that are elicited from real-world human-LLM interactions, as these are most relevant to studying the behavior of existing LLMs under realistic value conflicts. While past work has noted that LLMs and humans apply different values when interacting [1], the flexibility of our pipeline means that it could be used to evaluate LLM behavior under value sets that are more established as conflicting in existing philosophy, such as those from Moral Foundations Theory or Schwartz’s Theory of Basic Values.
>
> **Reliance on LLM-as-a-judge:** The review claims that all LLM-as-a-judge evaluation steps are completed without human verification or agreement check. However, as noted in Sections 3.2 and 3.3, we run extensive human validation both to validate our LLM-as-a-judge filters (Appendix D.1) and open-ended response evaluation (Appendix D.2). We find that our filtering has high precision and that our open-ended evaluation has high IAA when comparing both to annotator outputs.
>
> **Steerability Methodology:** We defined the steerability metric in Section 4.2 (L280-283) and gave examples of all steering prompts used in Appendix K. We have also revised our submission to include examples of successful cases of LLM steering, which can be found in the revised Appendix L.
>
> [1] “Values in the Wild: Discovering and Analyzing Values in Real-World Language Model Interactions”, Huang et al, 2025

---

### Official Review · Reviewer_ushm · 2025-11-01

**Soundness:** 3
**Presentation:** 3
**Contribution:** 3
**Rating:** 6
**Confidence:** 4

**Summary:**

The authors propose a method (CONFLICTSCOPE) to learn how LLMs prioritize competing values. They build dilemmas from a user‑specified value set, apply (LLM) filters, then run interactions with a simulated user. They use a judge model to map responses to actions and a Bradley–Terry model to form value rankings. The authros find that models shift from protective to personal values. They also show that including an explicit value hierarchy in the system prompt improves alignment with a target ranking.

**Strengths:**

+ Studying how models resolve value conflicts is an important problem (and is in contrast with existing work that focuses more on single values)

+ It's nice that they have some human validation experiments

+ Clear experimental results

**Weaknesses:**

- The labeling of scenario impossibility feels like it could be inconsistent; the filter description says it should be practically impossible to take both actions but the prompt asks the judge to return true if it is possible to take both actions (scenarios are retained only when all dims are labeled true?). This seems like it could result in both doable which seems to water down the dilemma definition.

- The study seems a bit limited by the single-turn aspect. In practice, many conflicts are negotiated via (multi-turn) interaction, like clarification, reframing, de-escalation, etc. It would be more realistic to study if the shift to personal values still holds when the model can interact over turns.

- The mapping V: D \times A \to A assumes unique representations and (strict?) dilemmas. A lot of conflicts allow for notions of good enough (satisficing), abstaining, compromise, etc. The formalism feels restrictive.

**Questions:**

- Can the authors clarify the boolean pass/fail rule for the scenario impossibility filter and show retained itesm that would have been excluded under the definition?

- There seems to be a disagreement between the steering effect definition divided by a(Rd,Rt) and Fig. 3's caption with correpsonds to dividing by 1-a(Rd,Rt). How would steering change if you report both normalizations?

- Can the authors replicate the environment-model swap results for Personal-Protective and ModelSpec, and not just HHH? I'm curious what the differences (if there are any) would tell us

---

> ### Author Response · Authors · 2025-11-20
>
> We thank the reviewer for their thoughtful feedback.
>
> **Scenario Impossibility Filtering**: To avoid double negatives that could lead to degraded classifier performance, we prompted our LLM-as-a-judge to return True if both actions in a scenario are possible before inverting the LLM-as-a-judge output to obtain our filter output. It was an oversight to not include this in our description of the scenario impossibility filter, and we have clarified this in Appendix C.3 of the revised submission. We have confirmed that the filter was working as intended, i.e. only passing scenarios where it is impossible to take both actions.
>
> **Multi-Turn Interactions:** We chose to focus on single-turn interactions as they were easier both to analyze and to validate in human study, while still being significantly more realistic than the multiple-choice evaluations used in previous work. However, we agree that it would be even better to study models in multi-turn interaction with a simulated user, and hope to pursue this in future work.
>
> **Restrictive Formalization:** While many real-world conflicts can be resolved by compromise, we chose to use the formalization given in Section 2 as it lets us measure LLM propensities under situations in which value conflict is most apparent. Framing conflict scenarios as pairwise comparisons between values allows us to create our aggregated value rankings and most closely mirrors the philosophical definition of moral dilemmas as cases where moral agents must select from multiple incompatible obligations. We agree that there is room for more nuance in studying LLM behavior under value conflicts and hope that future work can explore this further.
>
> **How are we normalizing results for steerability?:** The steerability metric should be $\frac{a(R_s, R_t) - a(R_d, R_t)}{1 - a(R_d, R_t)}$; we have corrected the typo in Section 4.2 to reflect this. Normalizing by $1 - a(R_d, R_t)$ rather than by $a(R_d, R_t)$ gives our steerability metric an upper bound of $1$, making it easier to aggregate results across different experiments without overrepresenting cases where a model’s starting alignment was low. For example, if $a(R_d, R_t) = 0.001$ and $a(R_s, R_t) = 0.1$, our normalization would give a value of 0.099, while normalizing by $a(R_d, R_t)$ would give us a value of 99. We view the upper bound as more important than the lower bound for normalization, as unnormalized steerability values are generally positive.
>
> **Do model preferences change across environments for other value sets?:** In Section 5.2 of the main paper, we show that models shift significantly in favor of personal values in personal-protective value conflicts in open-ended evaluation, as compared to multiple-choice evaluation. In Appendix I, we also show that this experimental result replicates over the HHH value set - all models shift significantly in favor of helpfulness (which is more personal) over harmlessness (which is more protective) when moving to open-ended evaluation. Similarly, higher-level ethical values in the ModelSpec set, such as fairness, are somewhat deprioritized in open-ended evaluation compared to multiple-choice.

---

### Author Response · Authors · 2025-11-20

Thank you to all reviewers for their insightful comments and feedback. We are encouraged that the reviewers found the study of how models resolve value conflicts to be an important (Ushm) and underexplored (bwD5) problem, and that we address this gap in existing alignment datasets in a way that is highly extensible (MtAF), generalizable, and potentially very useful (5iqJ). We are also pleased that the reviewers found our experimental methodology more reliable than past, multiple choice-based work (5iqJ), and our results revealing (bwD5) and interesting (5iqJ). Finally, we are encouraged that the reviewers appreciated our human validation of the evaluation pipeline and LLM-as-a-judge, with reviewer MtAF calling it “a welcome sight”. We address reviewer comments below, as well as in responses to individual reviewers, and have updated our submission to reflect all promised changes.

While reviewers generally appreciated our human validation of the open-ended LLM-as-a-judge, Reviewers MtAF and 5iqJ raised concerns about replicability and bias related to the usage of a closed API model as the judge. In response to these concerns, we have expanded our robustness analysis in Appendix G to also consider two open-weights models in the role of environment (simulated user + judge) model. Specifically, we consider Llama-3.1-70B-Instruct and Llama-3.1-8B-Instruct, which can both be run in an academic environment. We find moderate agreement between both models and GPT-4.1, the environment model that we use, with Cohen’s Kappa values of 0.506 and 0.449, respectively (compared to 0.56 between GPT-4.1 and Claude-3.5-Sonnet, as previously reported in Appendix G). When comparing across the four environment models, we found that changing environment models did not lead to any statistically significant changes in helpfulness-harmlessness preference rate across any of the target models evaluated. These results strengthen the claim made in the original Appendix G that our open-ended evaluation is relatively robust to environment model bias. We hope these results also make it easier for future work to replicate our usage of LLM-as-a-judge for evaluation.

Additionally, in response to comments left by Reviewer ushm, we have also fixed a typo in the steerability metric definition in Section 4.2 and added additional clarification about the scenario impossibility filter implementation in Appendix C.3. In response to comments left by Reviewer bwD5, we have added concrete examples of cases where adding a steering system prompt changed model behavior, which can be found in Appendix L of the revision.

We hope that these updates improve the quality and clarity of our work and look forward to further discussion of our work with the reviewers.

---

### Meta-Review · Area_Chair_9H2k · 2026-01-09

**Summary:**

Reviewers broadly agree that CONFLICTSCOPE targets an important and underexplored question—how LLMs resolve competing values—and that the paper delivers clear empirical findings (shift from protective to personal values in open-ended settings; steerability via explicit value hierarchies).
The main concerns affecting the decision are:
- potential ambiguity in the scenario impossibility filter and metric definitions
- heavy reliance on LLM-as-judge and reproducibility
- scope/realism limits (single-turn interactions; restrictive dilemma formalization)

**Reviewer Concerns:**

Addressed by rebuttal:
- Scenario impossibility filtering (ushm): Clarified the judge prompt and inversion logic; claim the filter only keeps scenarios where taking both actions is impossible.
- steering methodology clarity (ushm, bwD5): Fixed the normalization typo; pointed to metric definition and added prompt examples and qualitative cases in appendix.
- LLM-as-judge reliability, bias, reproducibility (bwD5, MtAF, 5iqJ): Emphasized human validation; added Llama-3.1 70B/8B with moderate agreement; reported preference rates stable across tested environment models.


Still outstanding:
- Single-turn evaluation (ushm, 5iqJ): No multi-turn experiments; key findings might differ under multi-step interactions.
- User study with user-supplied value sets (MtAF): No concrete user-study plan/results.
- Robustness (ushm): Not fully clear that environment-swap robustness is shown as comprehensively for Personal–Protective and ModelSpec as requested.

**Reviewer Scores:**

ushm (6): likely unchanged.
bwD5 (4): likely unchanged or change to 6. Human validation and clarified steering details should alleviate the strongest reliability concerns.
MtAF (6): likely unchanged.
5iqJ (8): likely unchanged.

---

### Decision · Program_Chairs · 2026-01-26

Accept (Poster)